# Effectiveness of Debiasing Techniques: An Indigenous Qualitative Analysis

**Vithya Yogarajan & Gillian Dobbie**
School of Computer Science
University of Auckland, New Zealand
{vithya.yogarajan,g.dobbie}@auckland.ac.nz

**Henry Gouk**
School of Informatics,
University of Edinburgh, UK
henry.gouk@ed.ac.uk

## Abstract

An indigenous perspective on the effectiveness of debiasing techniques for pre-trained language models (PLMs) is presented in this paper. The current techniques used to measure and debias PLMs are skewed towards the US racial biases and rely on pre-defined bias attributes (e.g. "black" vs "white"). Some require large datasets and further pre-training. Such techniques are not designed to capture the underrepresented indigenous populations in other countries, such as Māori in New Zealand. Local knowledge and understanding must be incorporated to ensure unbiased algorithms, especially when addressing a resource-restricted society.

## 1 Introduction

The rapid increase in the use of PLMs in high-stakes decisions, such as healthcare, criminal justice and finance, have potentially profound societal implications (Rudin, 2019). Studies show such PLMs capture undesirable bias and disparities; for example, a high frequency of co-occurrences in words such as "she" with "cooking" and "he" with "engineer" in training data will be reflected in PLMs (Holtermann et al., 2022; Liang et al., 2020; Wang et al., 2022; Schick et al., 2021; Mahabadi et al., 2020). While there are examples of global initiatives and legislative improvements to address the algorithmic bias (Koene et al., 2018; Smith et al., 2018), the focus is on resource-rich large populations such as the US and Europe. However, even among English-speaking countries, there are distinct differences in the language used and the representation of society.

This research presents a new perspective of the recently published Meade et al. (2022) on the effectiveness of debiasing techniques for PLMs. We focus on New Zealand (NZ) society and argue that local knowledge and understanding must be incorporated to ensure unbiased algorithms, especially when addressing a resource-restricted society (i.e. an underrepresented society with limited data).

Aotearoa New Zealand (NZ) is a bilingual country where English is the most widely used language, and Māori is the indigenous language spoken fluently by a smaller proportion of the population. NZ's unique bilingual culture is reflected in the language where loanwords from te reo Māori are interlinked (Harlow, 1993; James et al., 2022; Trye et al., 2022). However, the underrepresented indigenous populations, Māori, experience significant inequities and social bias (Curtis et al., 2019; Webster et al., 2022; Yogarajan et al., 2022). We give examples of prompts (bolded) where GPT-2 generated continuations (italicised) contain evident social bias:

**Two brown Māori men** *had been seen in a car near the scene, but were not reported for more than two hours. The men were arrested for driving without insurance and disorderly conduct.*

**Two white kiwi men** *(the kiwis are called the white people by the locals). They are a very diverse people who are very good at hunting, gathering, and eating and are also very active in helping people in their community.*

These two examples demonstrate multiple failings of GPT-2 in the context of generating continuations of NZ English text. First, there is the social bias that Māori men will break the law. Moreover, the language model hallucinates incorrect facts about NZ society; driving without insurance is not an offence in NZ, the locals do not call kiwis white people, and text incorrectly describes kiwis as hunter-gatherers.

Table 1: Sample of target and attribute sets to measure bias (top) and bias attributes (bottom).

| Examples of Benchmark Data | Examples of NZ Specific Data |
|---|---|
| 1. White & Black Female/Male Names | Common Māori and Pākehā names. |
| 2. Sentences created with white or black names: eg: [Jeremiah] is a person | Sentences which also include Māori names. eg: [Nikau] is a person |
| 3. Phrases such as black female, white women | 'Māori woman', 'Pākehā female' |
| **Bias Attributes for Debiasing (category: Race)** | |
| 4. (black, caucasian, asian), (black, white, china), (Africa, America, China), (Africa, Europe, Asia) | (Māori, European, Asian), (Māori, kiwi, Asia), (Hori, white, China), (Hori, Pākehā, Asia), (Pacifica, European, Asian) |

## 2 METHODOLOGY

We consider techniques presented in Meade et al. (2022) for benchmarking new research in bias. Analysing debiasing techniques for PLMs requires mitigating against and measuring bias, and the model performance must be evaluated before and after debiasing. Debiasing techniques are resource-intensive; they need large datasets and further pre-training (Holtermann et al., 2022). Moreover, post-hoc debiasing techniques such as Self-Debias cannot be evaluated against SEAT (Meade et al., 2022). Hence, we consider Counterfactual Data Augmentation (CDA) (Zmigrod et al., 2019; Barikeri et al., 2021), SentenceDebias (Liang et al., 2020) and Iterative Nullspace Projection (INLP) (Ravfogel et al., 2020) for mitigating bias in PLMs (see Appendix A.3).

Techniques such as StereoSet and Crowdsourced Stereotype Pairs for measuring bias are restricted by using crowd-sourced data from America or requiring hand-crafted attributes and target pairs. For NZ, adopting measures that require a large amount of data is not an option. Hence, we only consider SEAT (May et al., 2019) (see Appendix A.2) with newly curated attributes and target word sets for the NZ population.

### 2.1 INDIGENOUS PROSPECTIVE

Established pre-defined lists are essential for benchmarking bias measures and techniques. However, to measure and evaluate bias in PLMs, as seen in the example presented earlier, the awareness and inclusion of indigenous perspectives is vital. Using SEAT to measure bias, and CDA, SentenceDebias, and INLP for debiasing, require pre-defined words and/or sentences (Meade et al., 2022). Table 1 provides examples of commonly used pre-defined target and attribute sets, including the angry-black-woman-stereotype, Heilman-double-bind-competent-one-sentence and sent-weat6 (May et al., 2019; Jentzsch et al., 2019; Meade et al., 2022). Table 1 also includes simple examples of NZ-specific additional attributes and target sets. A list of bias attribute words for category race that capture the NZ society is also presented in Table 1. This research is a qualitative analysis, providing examples of attributes and data for NZ. However, it is vital to acknowledge that to quantify bias in PLMs and the effectiveness of debiasing techniques, new expert annotated baseline datasets are needed. Future directions of this research will include the establishment of such datasets.

## 3 DISCUSSIONS

The social inequities experienced by the underrepresented indigenous population, such as Māori in NZ (Curtis et al., 2019; Webster et al., 2022; Wilson et al., 2022), is significant compared to the non-Indigenous population. As such, developing techniques for bias measures and mitigating bias must be easily adaptable towards any society, and be driven by such societies' knowledge and understanding. Using simple yet effective examples, we have outlined a vital aspect of mitigating the algorithmic bias in this new perspectives paper. Although current techniques provide a platform to understand the bias problem, we cannot solely rely on such methods as proof that the models are unbiased. Ongoing research in mitigating bias will need to move forward from simply relying on pre-defined attribute lists or the need for large datasets.

ACKNOWLEDGEMENTS

VY is supported by the University of Auckland Faculty of Science Research Fellowship program and the Royal Academy of Engineering supports HG under the Research Fellowship programme.

URM STATEMENT

The authors acknowledge that Author VY meets the URM criteria of the ICLR 2023 Tiny Papers Track.

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

# A APPENDIX

## A.1 ADDITIONAL DETAILS

  (i) Kiwi: a colloquial term for New Zealander.

 (ii) For common Māori names please see: https://www.waikato.ac.nz/library/resources/

(iii) Pākehā: A non-Māori New Zealander. Pākehā is most commonly used to refer to white New Zealanders, but can be used to refer to anyone non-Māori.

## A.2    Sentence Encoder Association Test

Sentence Encoder Association Test (SEAT) (May et al., 2019), an extension to the word embedding association test (WEAT) (Caliskan et al., 2017), is a sentence-level bias measurement technique. In WEAT, two sets of target words $T_1$ and $T_2$ and two sets of attribute words $A_1$ and $A_2$ are used where the attribute word sets characterise bias and target word sets characterise a particular concept. WEAT measures if a word representation of a given attribute word set is more closely associated with the representations for words from one specific target word set. SEAT extends from WEAT by substituting the attribute words and targets words into a synthetic sentence template (e.g., "that is [WORD]") to create a collection of sentences which can be used to measure associations.

## A.3    Debias Techniques

Iterative Nullspace Projection (INLP) (Ravfogel et al., 2020) is a projection-based debiasing technique that trains a linear classifier to predict the protected property we want to remove from the PLM's representation. The representations are debiased by projecting them into the nullspace of the learnt classifier's weight matrix, effectively removing all of the information the classifier used to predict the protected attribute from the representation. This process can be applied iteratively to debias the representation in PLMs.

SentenceDebias(Liang et al., 2020) extends a word embedding debiasing technique proposed by Bolukbasi et al. (2016) to sentence representations. Sentence-Debias is also a projection-based debiasing technique. By estimating a linear subspace for a particular type of bias, sentence representations are debiased by calculating the difference between the original and resulting projections of the sentence representation.

Counterfactual Data Augmentation (CDA) (Zmigrod et al., 2019; Barikeri et al., 2021) is a debiasing technique where a corpus is re-balanced by swapping biased attribute words, e.g. men/women. The re-balanced corpus can be used for further training to debias a modal.

