# OpenReview forum: "Effectiveness of Debiasing Techniques: An Indigenous Qualitative Analysis"
_ICLR.cc/2023/TinyPapers — Submitted to Tiny Papers @ ICLR 2023_

### Official Review · Reviewer_ep7H · 2023-03-30

**Confidence:** 4

**Summary Of Contributions:**

The authors present a new perspective on debiasing pre-trained language models with respect to racial attributes. The discussion frames the indigineous communities in New Zealand. The authors present the problematic of current debiasing approaches and how they should be adapted to correctly assess racial bias in NZ society.

**Rating:**

High Potential (HP): a submission which meets the reviewing criteria and has potential to make an impact on the field

**Strengths And Weaknesses:**

*Strengths*
- The paper takes an interesting and useful approach to assess social bias.
- The paper is well written and meet the reviewing criteria: it is clear, there is appropiate discussion of relevant literature, and the claims made in the paper are justified by their findings.

*Weaknesses*
- The paper is reproducible. However, the lack of empirical excercises makes the title inadequate: since bias is not being measured, it is hard to tell the effectiveness of debiasing techniques without quantifying it. This is more a qualitative analysis, which is fine, but should be reflected in the title.


**Suggested Changes:**

- I would re-order the abstract to start presenting the problematic of what is currently being done, then continue with what is done in this paper. And then add a final sentence summarizing your findings. Also, in 'pre-defined bias attributes', do you mean 'racial attributes'?
- Footnote 2 show be next to the first appearance of the term (in the prompt 'Two white kiwi men').
- In section 2, the following claim should be rephrased as it seems to over-simplify the problem 'Analysing debiasing techniques for PLMs requires two aspects: mitigating against bias and measuring bias.' --> For instance, a good analysis would also require to measure model performance before and after debiasing, to ensure that the debiased model still performs correctly.

---

### Official Review · Reviewer_YHxE · 2023-03-30

**Confidence:** 3

**Summary Of Contributions:**

Authors have provided some insights on how pre-trained large language models(PLM) such as GPT-2, can have bias related to under represented indigenous societies. Authors have used biasness of PLM for New Zealanders and its' indigenous population Maori and then suggested most suitable techniques(from available technqiues) for debiasing without re-training.

**Rating:**

High Potential (HP): a submission which meets the reviewing criteria and has potential to make an impact on the field

**Strengths And Weaknesses:**

Strengths:
1. Clarity in the objective, easy to read and enough literature is provided by the author.
2. Authors have provided, an insight how currently all debiasing techniques are focused towards US racial biases however other under represented societies such as New Zealand indigenous population have been ignored.
3. They have provided which of the available debiasing techniques will be better for debiasing as data is very less for such indigenous populations.



**Suggested Changes:**

Paper is more of a qualitative analysis but it would be better if authors can report how does the suggested debiasing techniques perform for New Zealand indigenous group will be very useful. As it can help to address the same issues for other indigenous groups as well.

---

### Official Review · Reviewer_iua7 · 2023-04-03

**Confidence:** 1

**Summary Of Contributions:**

The paper talks about a very important problem in PLMs regarding bias in the data that it is trained on and highlights specifically for the case of indegenous community of NZ. The author highlights the techniques with relavant references as starting points for the debaising.

**Rating:**

Great Start (GS): a submission which meets some of the reviewing criteria but has room for improvement

**Strengths And Weaknesses:**

Strength: The problem is clearly explained along with relevant examples and techniques to bridge the bias gaps in PLMs\
Weakness: It is a good starting direction for debiasing but the effectiveness of the techniques along with the availability of additional data cannot be commented upon.

**Suggested Changes:**

Some sample cases where the application of such techniques worked would make it clear if there is a possibility of transferability.

---

### Author Response · Authors · 2023-05-30
**opt-in for archival**

We wish to opt-in for archival

---

### Comment · Area_Chair_rRYr · 2023-06-06
**Archival Criterion Check**

This work meets the threshold for archival, contents the URM statement and is deanonymized.

---

### Meta-Review · Area_Chair_rRYr · 2023-04-08

**Recommendation:** Invite to present
**Confidence:** 4

**Metareview:**

The authors investigate an important question in the AI community: debiasing. The paper provides a well-introduced background and discusses sufficient related works.

Strengths:

1. The problem is clearly explained with relevant examples and techniques for addressing bias gaps in pre-trained language models.
2. Clarity in the objective, easy to read, and a sufficient amount of literature is provided.
3. The paper highlights the focus on US racial biases and emphasizes the need to address underrepresented societies such as New Zealand's indigenous population.
4. It suggests suitable debiasing techniques for scarce data of indigenous populations.
5. The paper is well-written and meets the reviewing criteria.

Weaknesses:

1. The effectiveness of the techniques and the availability of additional data cannot be confirmed.
2. Lack of empirical exercises makes it difficult to assess the effectiveness of debiasing techniques without quantifying them.
3. The paper's title should reflect the qualitative analysis approach rather than suggesting a direct measurement of bias.

**Summary:**

The authors investigate biases in pre-trained language models, specifically targeting underrepresented indigenous societies like New Zealand's Maori population. They discuss the challenges of current debiasing approaches and suggest more suitable techniques for addressing these biases without retraining, emphasizing the importance of fairly representing indigenous communities in language models.

**Comments And Feedback To The Authors:**

You are addressing an important question in the AI community: debiasing. The paper is clear and reproducible, but it is still advised to revise the paper based on the reviewer's comments, such as providing sample cases for transferability of the debiasing techniques, including the performance of these techniques for the New Zealand indigenous group, reordering the abstract for clarity, and refining technical details.

**Reason For Not Giving A Higher Recommendation:**

The authors are advised to revise the paper by addressing the reviewer's comments, such as providing sample cases for transferability of the debiasing techniques, including the performance of these techniques for the New Zealand indigenous group, reordering the abstract for clarity, and refining technical details.

**Reason For Not Giving A Lower Recommendation:**

As the paper approaches the problem qualitatively rather than quantitatively, a lower score due to the absence of effectiveness comparisons for the debiasing techniques may not be justified.

---

### Decision · Program_Chairs · 2023-04-08

Invite to present